# Clusters of Comorbidities in the Short-Term Prognosis of Acute Heart Failure among Elderly Patients: A Retrospective Cohort Study

**DOI:** 10.3390/medicina58101394

**Published:** 2022-10-05

**Authors:** Lorenzo Falsetti, Giovanna Viticchi, Vincenzo Zaccone, Emanuele Guerrieri, Ilaria Diblasi, Luca Giuliani, Laura Giovenali, Linda Elena Gialluca Palma, Lucia Marconi, Margherita Mariottini, Agnese Fioranelli, Gianluca Moroncini, Adolfo Pansoni, Maurizio Burattini, Nicola Tarquinio

**Affiliations:** 1Department of Emergency Medicine, Internal and Sub-Intensive Medicine, Azienda Ospedaliero-Universitaria “Ospedali Riuniti”, 60100 Ancona, Italy; 2Department of Experimental and Clinical Medicine, Neurological Clinic, Azienda Ospedaliero-Universitaria “Ospedali Riuniti”, 60100 Ancona, Italy; 3Emergency Medicine Residency Program, Marche Polytechnic University, 60100 Ancona, Italy; 4Internal Medicine Department, Istituto Nazionale di Ricerca e Cura Anziani, INRCA-IRCCS, 60027 Ancona, Italy; 5Department of Experimental and Clinical Medicine, Clinica Medica, Azienda Ospedaliero-Universitaria “Ospedali Riuniti”, 60100 Ancona, Italy; 6Emergency Medicine Department, Istituto Nazionale di Ricerca e Cura Anziani, INRCA-IRCCS, 60027 Ancona, Italy

**Keywords:** acute heart failure, Charlson comorbidity index, comorbidities, elderly

## Abstract

*Background and Objectives:* Elderly patients affected by acute heart failure (AHF) often show different patterns of comorbidities. In this paper, we aimed to evaluate how chronic comorbidities cluster and which pattern of comorbidities is more strongly related to in-hospital death in AHF. *Materials and Methods:* All patients admitted for AHF to an Internal Medicine Department (01/2015–01/2019) were retrospectively evaluated; the main outcome of this study was in-hospital death during an admission for AHF; age, sex, the Charlson comorbidity index (CCI), and 17 different chronic pathologies were investigated; the association between the comorbidities was studied with Pearson’s bivariate test, considering a level of *p* ≤ 0.10 significant, and considering *p* < 0.05 strongly significant. Thus, we identified the clusters of comorbidities associated with the main outcome and tested the CCI and each cluster against in-hospital death with logistic regression analysis, assessing the accuracy of the prediction with ROC curve analysis. *Results:* A total of 459 consecutive patients (age: 83.9 ± 8.02 years; males: 56.6%). A total of 55 (12%) subjects reached the main outcome; the CCI and 16 clusters of comorbidities emerged as being associated with in-hospital death from AHF. Of these, CCI and six clusters showed an accurate prediction of in-hospital death. *Conclusions:* Both the CCI and specific clusters of comorbidities are associated with in-hospital death from AHF among elderly patients. Specific phenotypes show a greater association with a worse short-term prognosis than a more generic scale, such as the CCI.

## 1. Introduction

Heart failure (HF) is a global epidemic, often affecting elderly subjects afflicted by several comorbidities [1,2]. The interplay between HF and its associated disorders can worsen or complicate the clinical picture through several means, including pathology–pathology, drug–drug, and drug–pathology interactions. Several studies underlined increased morbidity, risk of rehospitalization, and mortality, both in acute decompensated HF (AHF) and chronic HF (CHF) of elderly subjects, especially when associated with comorbidity and multimorbidity [2,3]. Several predictive models for AHF survival have been proposed, and data resulting from a meta-analysis reported that the strongest predictors for AHF survival were age, sex, brain natriuretic peptide (BNP), serum creatinine, serum sodium, blood pressure, left ventricle (LV) function, New York Heart Association (NYHA) category, diabetes, body mass index, and exercise capacity [4]. The above-mentioned elements represent the features of specific disorders complicating AHF.

Age is a major determinant of complexity [4], and it should be considered as a common background for both chronic disease clustering, frailty, and worsening AHF prognosis. However, each associated disease has an incremental role in worsening AHF prognosis and quality of life.

A reduced exercise capacity can be observed in chronic obstructive lung disease (COPD), which often exacerbates AHF and represents an independent risk factor for death and hospitalization for HF [5]. Subjects with AHF overlapping COPD are often admitted with higher rates of LV systolic [6] and diastolic [7,8] dysfunction, increased BNP [9], lower blood pressure, and higher serum creatinine levels. Moreover, COPD subjects are often undertreated with ACE inhibitors, mineral-corticoid receptor antagonists (MRA), and beta-blockers.

Increased serum creatinine levels and electrolyte imbalances are common in chronic kidney disease (CKD), which often affects elderly, diabetic patients with HF [10], being associated with a more complicated presentation and an increased risk of death [11]. This group of subjects is also characterized by a lower blood pressure, an increased BNP, a higher NYHA category, and a more difficult in-hospital management. Moreover, loop diuretics are less effective in CKD, and dose escalation should be limited due to the risk of worsening renal failure. ACE inhibitors or MRA use is also limited by the risk of both hyperkaliemia and renal function deterioration.

Non-valvular atrial fibrillation (NVAF) worsens cardiac hemodynamics by reducing LV performances and is associated with an increased risk of death, hospitalization, and a worse quality of life, particularly in subjects with reduced ejection fraction (EF) [12,13].

Chronic anemia (CA) has been identified as an independent risk factor for mortality in AHF [14,15]. Patients with AHF co-affected by CA are usually older, with a lower blood pressure, higher serum creatinine, and increased BNP levels [15]. These subjects are commonly affected by several associated disorders, such as type 2 diabetes mellitus (T2DM), CKD, cardiovascular diseases (CVD), and COPD [16]. The causes of CA in HF can be found in the absolute or relative iron deficiency, impaired erythropoietin production due to CKD, reduced erythrogenesis caused by the renin–angiotensin blockade, hemodilution, and CA due to a chronic inflammation state, as in COPD. 

Cardiac comorbidities are common among patients affected by myelodysplastic syndromes (MDS) [17]. AHF associated with T2DM often show a phenotype characterized by hypertension, obesity, CKD, CA, and cardiovascular disease (CVD), which has already been associated with specific cardiovascular alterations, increased length of hospitalization, and mortality [18].

AHF can be associated, especially in the elderly, with severe concurring clinical events, such as, for example, ACS or the Takotsubo syndrome [19], but also acute valvular diseases or infective pathologies. These associations impact significantly on the patients’ survival, which can be further reduced in the presence of common comorbidities, such as NVAF or cancer [20,21]. To focus mainly on the impact of the burden of comorbidities on AHF prognosis, we excluded patients affected by these associations, since their short-term survival and management significantly differs from the other AHF patients, as already elucidated in previous studies exploring the same topic [22].

The role of generic comorbidity scores, such as the Charlson comorbidity index (CCI), has already been studied in HF. CCI has been associated with an increased mortality in the first year after the first AHF episode [22,23], while it was not able to predict CHF prognosis [24]. The advantages of CCI are mainly related to its large validation in different clinical settings and to its broad coverage of the most common comorbidities. However, some specific conditions, already associated with worse outcomes in AHF, such as non-valvular atrial fibrillation (NVAF) or CA, are not included in CCI [25].

In this work, we aimed to evaluate whether the different clusters of comorbidities can predict short-term prognosis in a population of elderly patients admitted for AHF, considering both de novo AHF (first AHF episodes) and acutely decompensated heart failure (ADHF). We also evaluated CCI performance in predicting in-hospital death from AHF and compared the performance of CCI with the phenotypes of comorbidities observed in our population. 

## 2. Materials and Methods

We retrospectively enrolled all the consecutive patients admitted to the Internal Medicine Department of INRCA-IRCSS of Osimo (Ancona) from the Emergency Department of the same hospital for AHF over 4 years (from 01/2015 to 01/2019), aged ≥65 years. The AHF diagnosis was made by the attending physician, according to the guidelines relevant at the moment of study [26]. 

We investigated age, sex, BNP at admission, BNP at discharge, echocardiography-assessed cardiac EF at admission, the number of comorbidities, the number of drugs taken before admission, the days of hospitalization, and the presence of 17 different comorbidities. For each subject, we calculated CCI according to its original definition [25]. The main outcome of the study was defined as in-hospital mortality following admission for AHF. 

### 2.1. Ethical Approval and Data Availability Statement

The ethics committee reviewed the protocol and approved the study (Comitato Etico INRCA, Protocols: CE-INRCA-19012; 33926/20-CE and 411/DGEN). All the participants and/or caregivers gave their written informed consent. All the subjects were treated according to the Declaration of Helsinki. The data that support the findings of this study are available from the corresponding author upon reasonable request.

### 2.2. Definition of Comorbidities

COPD was defined by a clinical history supported by previous suggestive spirometric findings; obstructive sleep apnea syndrome (OSAS) was established by the presence of a previous diagnosis supported by polysomnographic data interpreted by an expert neurologist; CA was ascertained if the hemoglobin levels were lower than 13 g/dl for men or 12 g/dl for women for at least 6 months before admission without clinical or instrumental findings of bleeding; the presence of dementia (DEM) was evaluated according to the patient’s clinical history in the presence of a previous neuropsychologic evaluation. An acute deterioration of cognitive function was not considered. Hypertension (HYP) was appraised in the subject’s clinical history and by the chronic use of anti-hypertensive medications given for this specific application. An in-hospital finding of increased blood pressure levels was not considered. Dyslipidemia (DYS) was defined by the patient’s history or lipid-lowering-agents use for this specific purpose. CKD was defined as the persistence of kidney function abnormalities for at least 3 months of clinical stability before hospital admission [27]. Moderate-to-severe CKD was defined as a stable reduction in the glomerular filtration rate (eVFG) ≤ 60 mL/min estimated with the CKD-EPI formula, in a period of clinical stability before the hospital admission for AHF. The finding of an acute or subacute reduction in eVFG at admission was not deemed sufficient for this diagnosis. Patients with end-stage renal disease in dialysis were excluded from the study. T2DM was considered in the presence of a previous T2DM diagnosis or by the chronic use of anti-diabetic therapies. We also evaluated the presence of end-organ involvement to correctly assess CCI. Patients affected by type 1 diabetes mellitus were excluded from this analysis. Permanent NVAF (NVAF) was confirmed by the patient’s history and electrocardiographic findings. Patients with persistent or paroxysmal NVAF were excluded from this analysis. The presence of peripheral artery disease (PAD) was established by a history of supra-aortic, aortic, or lower-limb atherosclerotic vascular disease confirmed by previous ultrasonographic, angiographic, or CT-angiographic examinations. Cerebrovascular disease (CED) was confirmed by a history of previous stroke or TIA, while coronary artery disease (CAD) was defined by a history of previous acute myocardial infarction, a previous cardiac revascularization procedure, or by a coronary angiographic finding of coronary atherosclerosis. We considered in the hematologic disorders (HD) category all the chronic hematologic diseases (myelodysplastic syndromes, polycythemia vera, essential thrombocythemia), excluding lymphoma, leukemia, and multiple myeloma, which were considered neoplastic disorders. Active cancer (AC) was ascertained by the presence of any solid or liquid cancer with no instrumental signs of remission at admission. The presence of positive lymph nodes or metastases was recorded to correctly assess CCI. Collagen disorders (CD) were considered if a diagnosis of lupus erythematosus, rheumatoid arthritis, Sjogren syndrome, or systemic sclerosis/scleroderma was present at the moment of hospital admission. Thyroid disorders (THYR) were assessed at admission and defined as the presence of thyroid gland dysfunctions before the acute event. We did not consider thyroid function abnormalities diagnosed during hospitalization. Chronic infectious diseases (C_INF) were defined as HIV/AIDS, chronic HBV infection, and chronic HCV infection.

### 2.3. Inclusion and Exclusion Criteria

All subjects aged 65 years old or older with AHF and admitted from the ED in the Internal Medicine Department were enrolled. We excluded (a) subjects younger than 65 years old and (b) all patients admitted for AHF among whom we observed a second acute related or unrelated condition that could have significantly modified the prognosis. Per protocol, we excluded patients admitted for AHF when it was associated with (i) acute coronary syndromes (ACS) or other acute coronaric diseases, such as the Takotsubo syndrome, (ii) acute valvular diseases, (iii) acute infective events, including pneumonia, sepsis, or septic shock, (iv) terminal conditions in palliative care, and (v) end-stage renal disease requiring dialysis. We also excluded patients with an incomplete medical history, whose electronic medical records did not allow us to calculate CCI, with incomplete follow-up, or subjects admitted from departments other than the ED.

All the enrolled patients were subdivided, according to their clinical history, into de novo AHF (patients without a clinical history of AHF or CHF) and acutely decompensated heart failure (ADHF, patients with a clinical history of previous AHF or CHF).

### 2.4. Statistical Analysis

In-hospital mortality, sex, and comorbidities were synthesized as dichotomous variables. Age, admission BNP, number of drugs used at admission, and days of hospitalization were collected as continuous variables. CCI was calculated according to its original definition [25] and collected as a continuous variable. We also generated a second variable recording CCI into four quartiles. 

To identify the clusters of chronic pathologies in this cohort, we adopted two-tailed Pearson’s bivariate test. We considered two disorders as being associated when Pearson’s statistic emerged as significant at a level of *p* ≤ 0.10. From the results of Pearson’s bivariate test, we selected “groups” of ≥2 significantly associated comorbidities or “clusters”.

The covariate selection for the multivariate models (logistic regression analysis and Cox proportional hazards model) was performed with Pearson’s bivariate test, selecting the features associated with the main outcome variable at a level of significance of *p* ≤ 0.10.

To evaluate the association between CCI and the main outcome in this sample, we prepared a binary logistic regression model considering in-hospital mortality as the dependent variable, CCI as the main predictor, and the covariates selected with Pearson’s test. Age was not included as a covariate in this model, since it was already considered in CCI. The individual predicted probabilities of this model were saved at the end of the binary logistic regression procedure and used to generate the ROC curve.

We considered for further assessments only the clusters whose binary logistic regression analysis model emerged as significant at a level of *p* ≤ 0.05 and whose uncorrected ROC curve had an AUC ≥ 0.65. The selected phenotypes were used to generate the complete models, which considered in-hospital mortality as the main outcome, the associated pathologies, and the covariates selected with Pearson’s test, including age. The individual predicted probabilities of each model were saved at the end of the binary logistic regression procedure and used to generate the ROC curves. We compared the performance of the ROC curve of each cluster with the other clusters and with the ROC curve derived from the CCI model, adopting the DeLong method.

Lastly, we ran a Cox proportional hazards model considering the main outcome as the dependent variable, days of admission as the time variable, the four CCI quartiles as the main independent variables, and the covariates selected with Pearson’s test, excluding age, since it was already considered in the CCI. 

## 3. Results

From an initial sample of 512 patients, we excluded 15 subjects due to ACS, 36 due to concurrence of acute infective events, and 2 due to acute valvular diseases, obtaining a final sample of 459 consecutive subjects. The baseline characteristics of the cohort are synthesized in Table 1. The main outcome (in-hospital death following admission for AHF) was reached in 55 patients (11.90%).

The CCI emerged as being significantly associated with the main outcome, with an AUC of 0.69 (95% CI: 0.58–0.75; *p* = 0.0001) in the uncorrected model, which increased in the corrected model, adjusting for sex, number of drugs used, admission BNP, and days of hospitalization (Table 2). 

Pearson’s bivariate analysis identified 17 different clusters of comorbidities, as shown in Figure 1. The binary logistic and ROC curve analyses underlined those 16 phenotypes were significantly associated with in-hospital mortality for AHF (Figure 1), but only 6 of them had a clinically acceptable AUC, defined by AUC values ≥ 0.65. After adjustment for age, sex, admission BNP, days of hospitalization, and number of drugs in the logistic regression analysis, we observed that these six phenotypes had similar performances in predicting the main outcome, as shown in Table 2. 

The phenotypes with a ROC curve ≥ 0.65 were phenotype 4 (DEM, CA, DYS, NVAF, CD), phenotype 6 (DYS, OSAS, DEM, HYP, T2DM, PAD, CED, CAD, AC, and C_INF), phenotype 9 (NVAF, DEM, T2DM, CAD, HD), phenotype 10 (PAD, COPD, CA, HYP, DYS, CKD, CAD), phenotype 11 (CED, OSAS, DYS, CAD, AC, HD), phenotype 12 (CAD, COPD, OSAS, DYS, CKD, NVAF, PAD, CED, THYR, C_INF), and phenotype 14 (HD, HYP, NVAF, CED, AC). We did not observe a statistically significant difference between the ROC curves of the six selected clusters, while the CCI showed significantly lower performances than the selected clusters, particularly when compared with phenotypes 6, 9, 11, and 12, as shown in Figure 2 and Table 3.

In the Cox regression model, considering sex, number of drugs, and BNP at admission as covariates, we observed a significantly increased risk of in-hospital death from the first to the fourth CCI quartile, as shown in Table 4.

## 4. Discussion

Patients admitted for AHF represent a markedly heterogeneous group who differ by the etiology, severity of clinical presentation, and associated diseases. Our cohort was characterized by elderly patients burdened by several associated pathologies and treated with several drugs before admission, confirming the association between HF, significant morbidity, and polypharmacy observed in previous studies [28,29]. Non-cardiac comorbidities are present in more than 30% of AHF patients evaluated in large registries and trials [16]. The ADHERE, OPTIMIZE-HF, and EHFS-II studies show a large prevalence of COPD, CKD, T2DM, CA, depression, and liver disease in a population aged between 70 and 75 years [11,30,31]. The presence of non-cardiac comorbidities affects both new hospitalizations and rehospitalization for HF [32,33]. 

Even the presence of a single comorbidity complicates the clinical course of AHF. Patients with COPD can be more difficult to diagnose and treat due to the presence of respiratory insufficiency, while patients with CKD often face lower response rates to diuretics, and decongestion is often complicated by a worsening of the renal function, which reduces the use of ACE-I, ARB, and MRA [16]. When comorbidities cluster, the patient becomes even more complex and difficult to manage, especially in the most acute phase of the disease. Moreover, patients affected by “comorbid” AHF are often managed in Internal Medicine Departments, with age, demographic factors, and non-cardiac comorbidities being well-established barriers to specialty referral [34]. 

In our study, we observed a mean of four comorbidities and a median of seven drugs, suggesting a high clinical complexity of this typology of subjects. AHF in the elderly represents a diagnostic and therapeutic challenge, especially when associated with multiple comorbidities and a subsequent state of polypharmacy. In this population, survival is also affected by age and pathology–pathology interactions. Drug–pathology and drug–drug interactions must also be considered when treating this specific group of patients.

In our cohort, we were able to identify six different phenotypes characterized by the clustering of chronic pathologies and to evaluate their association with short-term AHF prognosis. This analysis allowed us to underline which associations of pathologies, or phenotypes, were significantly associated with the main outcome. We observed that vascular, chest, and metabolic diseases often cluster in the same individual to generate a state of high clinical complexity and a worse short-term outcome. 

Two of the phenotypes identified with our methods were characterized by metabolic (phenotype 6), vascular (phenotype 10), and metabolic–vascular (phenotype 12) diseases, while the other remaining clusters were depicted by the association of NVAF and dementia in connection with metabolic (phenotype 4), ischemic (phenotype 9), and neoplastic (phenotype 14) diseases. Of note, NVAF is highly prevalent and correlated to a worse prognosis in several phenotypes of this critical illness, as already observed in other studies [35]. The association of this arrhythmia with dementia, particularly vascular dementia, or Alzheimer’s disease, is well known [36]. Our results are in line with recent studies and confirm the importance of the clinical phenotype in the determination of short-term prognosis of AHF/ADHF [37,38].

We also evaluated the role of a more generic score, the Charlson comorbidity index, in the setting of AHF and compared its performance with the AUC resulting from the analysis of each specific cluster. CCI represents a validated assessment of the patient’s comorbidity status and has a definite role in predicting long-term mortality [25,39]. Several authors already studied its performances in different acute pathologies, such as stroke [40], acute myocardial infarction [41,42], and AHF [22,23], suggesting that the coexistence of advanced age and associated pathologies could represent a strong risk factor in worsening the patients’ short-term prognosis during the acute phase of a disease. 

Several predictive models, such as the EHMRG score, have been suggested to assess short-term AHF prognosis since the patient’s arrival in the ED. These models are largely validated and should be preferred to stratify the 7-day mortality risk over other approaches, even in a geriatric patient [43,44]. However, our study suggests that applying the clinical method and correctly assessing the patient’s burden of comorbidities could be useful for the practicing clinician to evaluate the patient’s complexity and assess their short-term mortality risk according to their history.

In our study, we also confirm the prognostic role of CCI in AHF short-term mortality, lower quartiles of CCI being associated with a lower risk of in-hospital death. However, when comparing specific clusters of chronic diseases with a more generic but validated score, we observed that six specific phenotypes were able to outperform the CCI in the prognostic evaluation of elderly patients affected by AHF. 

These observations lead to some considerations that can be useful for the physician. First, the CCI should be evaluated at admission of elderly subjects with AHF to weigh rapidly the individual complexity and to stratify the subject’s short- and long-term prognosis. CCI is largely used in the evaluation of geriatric patients; moreover, it is a validated and extensive score that gives the physician several hints as to the patient’s status. 

Second, our results suggest that an accurate investigation on the patient’s history, focused mainly on vascular and metabolic comorbidities and their interaction, is useful to identify the subjects at the highest risk of clinical deterioration, underlining the importance of applying a good clinical method. 

The major limitations of this work are mainly related to its monocentric and retrospective design. For this reason, this must be considered as a pilot study, and the results must be considered as preliminary, requiring validation in larger prospective cohorts.

## Figures and Tables

**Figure 1 medicina-58-01394-f001:**
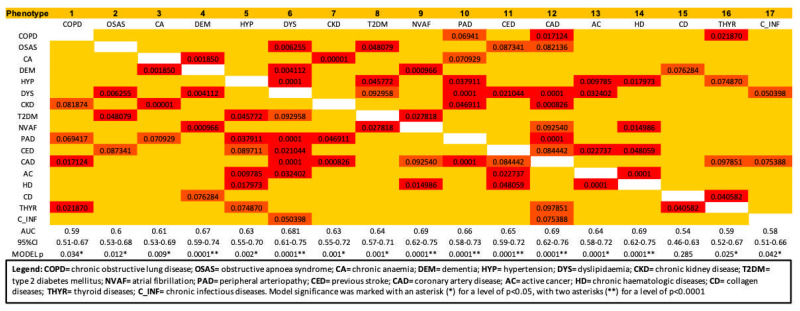
Heat map of Pearson’s bivariate test, underlining the clusters of comorbidities and the strength of association (in red: significant association between comorbidities at 0.05 level, in orange: significant association between comorbidities at 0.10 level). The row named “ROC” is used to show area under the ROC curve defining the accuracy of the prediction of the cluster (defined by the column) for in-hospital mortality. The row named “95% CI” refers to the 95% confidence interval of the ROC curve, while the row named “MODEL p” refers to the significance of the model adopted for ROC curve analysis.

**Figure 2 medicina-58-01394-f002:**
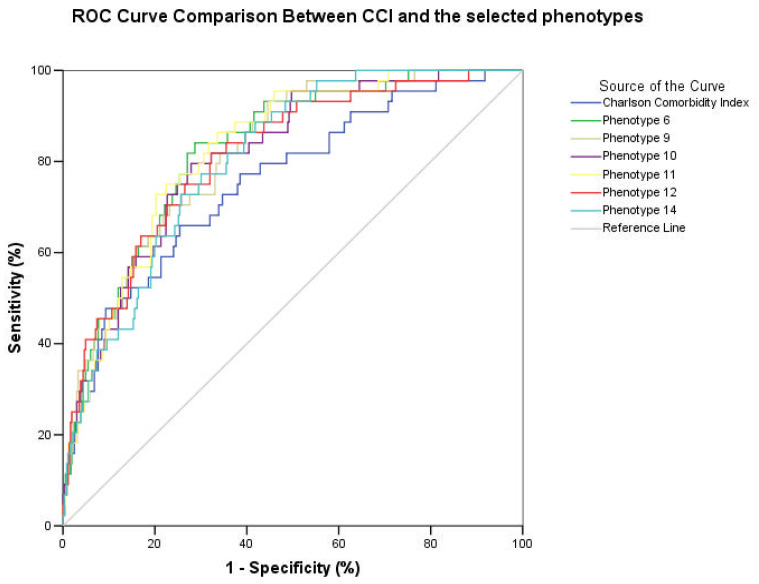
ROC curves showing the accuracy of the prediction of in-hospital death of the selected phenotypes and Charlson comorbidity index. A detailed comparison of each phenotype with CCI can be found in Table 4.

**Table 1 medicina-58-01394-t001:** Baseline characteristics of the enrolled cohort.

In-hospital death (*n*, %)	55 (11.90%)
Days of admission (median, [IQR])	10 [6]
Type of AHF de novo AHF ADHF	179 (39.0%) 280 (61.0%)
Male sex (*n*, %)	260 (56.60%)
Age (mean, ±SD), years	83.98 (±8.02)
Charlson comorbidity index (median, [IQR])	6 [2]
Drugs taken at admission (median, [IQR])	7 [5]
General characteristics	
BNP at admission (mean, ±SD), ng/mL	977.15 (±212.6)
BNP at discharge (mean, ±SD), ng/mL	737.82 (±115.8)
Ejection Fraction (mean, ±SD), %	46 (±13.1)
Comorbidities	
Number of comorbidities (mean, ±SD)	4.44 (±1.89)
COPD (*n*, %)	82 (17.90%)
OSAS (*n*, %)	8 (1.70%)
Chronic Anemia (*n*, %)	154 (33.60%)
Dementia (*n*, %)	71 (15.50%)
Hypertension (*n*, %)	332 (72.30%)
Dyslipidemia (*n*, %)	186 (40.50%)
Chronic Kidney Disease (*n*, %)	212 (46.20%)
Diabetes (*n*, %)	140 (30.40%)
Atrial Fibrillation (*n*, %)	249 (54.10%)
Peripheral Artery Disease (*n*, %)	90 (19.60%)
Previous Stroke or TIA (*n*, %)	72 (15.70%)
Previous Acute Myocardial Infarction (*n*, %)	110 (24.00%)
Active Cancer (*n*, %)	97 (21.10%)
Hematologic Pathologies (*n*, %)	50 (10.90%)
Connective Tissue Diseases (*n*, %)	38 (8.30%)
Thyroid Diseases (*n*, %)	78 (17.00%)
Chronic Infectious Diseases (*n*, %)	17 (3.70%)

Legend: AHF = acute heart failure; ADHF = acute decompensated heart failure; BNP = brain-derived natriuretic peptide; COPD = chronic obstructive lung disease; IQR = interquartile range; OSAS = obstructive sleep apnea syndrome; SD = standard deviation; TIA = transient ischemic attack.

**Table 2 medicina-58-01394-t002:** ROC curves of specific phenotypes and Charlson comorbidity index for in-hospital death.

	Uncorrected Model		*p*	Corrected Model		*p*
	AUC	95% CI		AUC	95% CI	
Phenotype 4	0.67	0.59–0.74	0.0001	0.80	0.73–0.87	0.0001
Phenotype 6	0.68	0.61–0.75	0.0001	0.83	0.77–0.88	0.0001
Phenotype 9	0.69	0.62–0.75	0.0001	0.82	0.76–0.88	0.0001
Phenotype 10	0.66	0.58–0.73	0.0001	0.81	0.75–0.87	0.0001
Phenotype 11	0.65	0.59–0.72	0.0001	0.83	0.77–0.88	0.0001
Phenotype 12	0.69	0.62–0.76	0.0001	0.81	0.75–0.88	0.0001
Phenotype 14	0.69	0.58–0.72	0.0001	0.81	0.75–0.86	0.0001
CCI	0.69	0.58–0.75	0.0001	0.73	0.64–0.82	0.0001

Legend: AUC = Area Under the Curve; CI = Confidence Interval; CCI = Charlson Comorbidity Index.

**Table 3 medicina-58-01394-t003:** Comparison between specific phenotypes and Charlson comorbidity index.

	AUC (95% CI)	Delta_AUC_	*p* (vs. CCI)
Phenotype 4	0.80 (0.73–0.87)	7.09%	0.114
Phenotype 6	0.83 (0.77–0.88)	9.87%	0.030 (*)
Phenotype 9	0.82 (0.76–0.88)	9.28%	0.036 (*)
Phenotype 10	0.81 (0.75–0.87)	8.32%	0.089
Phenotype 11	0.83 (0.77–0.88)	9.72%	0.027 (*)
Phenotype 12	0.81 (0.75–0.88)	8.38%	0.048 (*)
Phenotype 14	0.81 (0.75–0.86)	7.83%	0.063

Legend: AUC = Area Under the Curve; CI = Confidence Interval; Delta_AUC_ = difference between AUC curves; CCI = Charlson Comorbidity Index; significant differences were marked with an asterisk (*)

**Table 4 medicina-58-01394-t004:** Cox regression analysis for in-hospital death following admission for AHF7.

	HR	95% CI	*p*
CCI Q1	(ref.)	-	-
CCI Q2	5.042	1.133–22.442	0.034
CCI Q3	5.336	1.087–26.196	0.039
CCI Q4	8.509	1.932–37.465	0.005
Sex	1.307	0.691–2.471	0.411
Number of drugs at admission	0.910	0.825–1.004	0.061
BNP at admission	1.005	1.001–1.010	0.000

Legend: BNP = Brain-Derived Natriuretic Peptide; CI = Confidence Interval; CCI = Charlson Comorbidity Index; HR = Hazard Ratio; Q1–Q4 = CCI Quartiles (1–4).

## Data Availability

The data that support the findings of this study are available from the corresponding author upon reasonable request.

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
