# Peer review of "Clusters of Comorbidities in the Short-Term Prognosis of Acute Heart Failure among Elderly Patients: A Retrospective Cohort Study"

_medicina, 2022, doi:10.3390/medicina58101394_

Round 1

Reviewer 1 Report

Falsetti et al. present their data on AHF and cluster analysis in  a retrospective regisry over several years. I have the following concerns

1-Abstract: data need to be better elobarated. What is the outcome? Please summarize better the results in the abstract

2-Introduction

It makes sense to mention possible diseases of acute heart failure and chronic heart failure. Of note, elederly patients may present frequently acute heart failure e.g. Takotsubo syndrome (PMID: 27341847). Patients may suffer relevant comorbidities e.g. atrial fibrillation and cancer (PMID: 27702871, PMID: 28318661)

Methods:

Did you include patients suffering from Takotsubo syndrome?

Results:

You need to better describe your findings. What was the underlying disease of AHF?

What is the main outcome? Please describe better in the text the different phenotypes

Discussion:

the discussion part needs a better analysis of published data and the add-on of your results.

Reviewer 2 Report

The authors examined chronic comorbidities clustering relating to in-hospital deaths of patients presenting with acute heart failure (AHF).  They identified 16 clusters.  Specific phenotypes showed a greater association.  The results support the conclusion.

No further suggestion.  

Reviewer 3 Report

In this manuscript, the authors investigated the significance of Charlson Comorbidity Index (CCI) should be evaluated at the admission of elderly patients with acute heart failure to assess rapidly the individual complexity and to stratify the subject’s short- and long-term prognosis. The experiments and methods are sound, and the data presented are convincing to support the authors’ conclusions. However, there are major concerns that reduce enthusiasm for the manuscript. 

1. The author should state the advantage of the Charlson Comorbidity Index clearly in the introduction part.

2. In the methods part, the enrollment criteria made me confused. “The criteria for enrollment included an absence of a history of diabetes mellitus”?

3. The mechanism why the four CCI quartiles as the main independent variables, and sex, number of drugs, and BNP at admission can represent as covariates should be discussed with Cox Regression Analysis in the manuscript. However, the relations between Table 3 and Figure 1 would not be powerful enough to conclude that directly. The authors should explain more.

4. Please address this in more detail in your revised manuscript and explain clearly, why your new finding is superior to other approaches. If, in the meantime, you have collected additional experimental evidence, it would be very helpful if you included that into your revision.

Minor points 

1. The illustration in Figure 2 should also be clear. 

2. It would benefit from significant content and grammatical editing throughout. There are numerous examples, but to illustrate a few in the abstract and introduction. This is a significant issue that must be corrected throughout the paper. 

Round 2

Reviewer 1 Report

The paper is much improved. Please check the whole paper for typical and grammatical errors.

Please discuss age-variation in heart failure.

Author Response

Q1. The paper is much improved. Please check the whole paper for typical and grammatical errors.

A1. Thank you for this overall appreciative question. In the revised version (R2) we have  corrected the typos and improved the English form

Q2. Please discuss age-variation in heart failure.

A2. We thank the reviewer for this suggestion. In the revised version we have added a sentence in the introduction briefly discussing the role of age in HF